# An Hsp90 co-chaperone protein in yeast is functionally replaced by site-specific posttranslational modification in humans

Abbey D. Zuehlke[1], Michael Reidy[2], Coney Lin[2], Paul LaPointe[3], Sarah Alsomairy[1], D. Joshua Lee[1], Genesis M. Rivera-Marquez[1], Kristin Beebe[1], Thomas Prince[1], Sunmin Lee[4], Jane B. Trepel[4], Wanping Xu[1], Jill Johnson[5], Daniel Masison[2] & Len Neckers[1]

Heat shock protein 90 (Hsp90) is an essential eukaryotic molecular chaperone. To properly chaperone its clientele, Hsp90 proceeds through an ATP-dependent conformational cycle influenced by posttranslational modifications (PTMs) and assisted by a number of co-chaperone proteins. Although Hsp90 conformational changes in solution have been well-studied, regulation of these complex dynamics in cells remains unclear. Phosphorylation of human Hsp90α at the highly conserved tyrosine 627 has previously been reported to reduce client interaction and Aha1 binding. Here we report that these effects are due to a long-range conformational impact inhibiting Hsp90α N-domain dimerization and involving a region of the middle domain/carboxy-terminal domain interface previously suggested to be a substrate binding site. Although Y627 is not phosphorylated in yeast, we demonstrate that the non-conserved yeast co-chaperone, Hch1, similarly affects yeast Hsp90 (Hsp82) conformation and function, raising the possibility that appearance of this PTM in higher eukaryotes represents an evolutionary substitution for *HCH1*.

[1] Urologic Oncologic Branch, Center for Cancer Research, National Cancer Institute, 9000 Rockville Pike, Bethesda, Maryland 20892, USA. [2] Laboratory of Biochemistry and Genetics, National Institute of Diabetes and Digestive and Kidney Diseases, National Institutes of Health, Building 8, Room 225, 8 Center Drive, Bethesda, Maryland 20892, USA. [3] Department of Cell Biology, Faculty of Medicine and Dentistry, University of Alberta, Edmonton, Alberta, Canada T6G 2H7. [4] Developmental Therapeutics Branch, Center for Cancer Research, National Cancer Institute, 9000 Rockville Pike, Bethesda, Maryland 20892, USA. [5] Department of Biological Sciences and the Center for Reproductive Biology, University of Idaho, Moscow, Idaho 83844, USA. Correspondence and requests for materials should be addressed to L.N. (email: neckers@nih.gov).

The homodimeric molecular chaperone Hsp90 regulates the activity and stability of ∼10% of cellular proteins, many of which are required for signalling and growth[1,2]. Proper chaperoning of Hsp90 clientele requires an ATP-influenced conformational cycle. Throughout this cycle, helper proteins, termed co-chaperones, interact with Hsp90 to assist in client interactions, conformational dynamics and ATP hydrolysis[3–5]. In the absence of nucleotide, Hsp90 primarily populates an 'open', N-domain undimerized conformation. This conformation is stabilized by the co-chaperones Hop (ySti1) and Cdc37 for client loading[6–10].

On binding nucleotide, Hsp90 proceeds into a 'closed' conformation characterized by transient dimerization of its N-terminal domains and strengthened client interaction[11]. Hsp90 in its closed conformation can be found in association with the co-chaperones Aha1 (closed state 1) or p23 (ySba1; closed state 2). Aha1 interacts with Hsp90 to accelerate ATP hydrolysis[12–15], while p23 stabilizes the closed conformation and increases client-Hsp90 interaction time[5,16–19]. Following ATP hydrolysis, Hsp90 releases the client and proceeds back into an open conformation. In eukaryotes, posttranslational modification (PTM) represents an additional regulatory mechanism underlying the Hsp90 ATPase cycle, and in human cells phosphorylation of Hsp90 drives co-chaperone and client recruitment and release[20].

Despite a substantial understanding of Hsp90 conformational changes in solution, factors influencing Hsp90 conformational state and client interactions in cells are less well understood. Recently, a region near the Hsp90 middle domain/C-terminal domain junction was shown to be necessary for proper client chaperoning by yeast Hsp90 (ref. 21). While amino acid mutation in this region strongly affects binding of an artificial substrate to bacterial Hsp90, similar mutations in the yeast chaperone appear to have a more indirect effect on client remodelling since their impact on client binding is minimal[21]. In this study, we provide evidence that the strongest mutation in this region of yeast Hsp90 (Hsp82), W585T, disrupts client remodelling coincident with stabilizing yHsp90 in an open conformation, even in the presence of non-hydrolyzable ATP. Further, we show that the non-conserved yeast co-chaperone Hch1 exacerbates the growth, drug sensitivity and client effects of Hsp82-W585T by also stabilizing the open conformation. Finally, we propose that the Hsp82-Y606E phosphomimetic mutation in yeast (equivalent to the previously identified Hsp90-Y627E in human cells[20]) has similar effects on growth, Hsp90 conformation and client activity as does expression of HCH1. Although this residue is not phosphorylated in yeast[22], Hsp90-Y627 phosphorylation and its phosphomimetic mutation have both been associated with client dissociation and loss of Aha1 binding in human cells[20]. Thus, our current data are consistent with the possibility that this PTM may serve a similar function as Hch1 in higher eukaryotes.

## Results

### Loss of HCH1 relieves the Hsp82-W585T growth phenotype.
Loss of HCH1 relieves the growth defect caused by the Hsp82-A587T mutation[23]. Since this amino acid is adjacent to the C-terminal region of Hsp90, whose mutation causes defects in yeast growth and client remodelling (Fig. 1a)[21], we asked whether Hch1 may similarly influence these phenotypes. First, we assessed the impact on growth of HCH1 expression in combination with the previously identified Hsp82-W585T mutation[21]. HSP82 WT or W585T were transformed and expressed as the only Hsp90 in otherwise WT or hch1Δ cells. Indeed, deletion of HCH1 relieved the negative growth phenotype associated with the Hsp82-W585T mutation at all tested temperatures (Fig. 1b). Consistent

with these observations, overexpression of HCH1, through use of a GPD-HCH1 construct, exacerbated the Hsp82-W585T growth phenotype (Fig. 1c). Using the GPD-HCH1 construct, we sought to determine whether Hch1 interaction with Hsp90 was necessary for its impact on the W585T phenotype. Indeed, mutation of Hch1 in the position required for Hsp90 interaction (D53 to N) relieved the Hsp82-W585T growth defect (Supplementary Fig. 1)[24,25].

### Hch1 reduces client accumulation and activity.
The Hsp82-W585T mutation was recently shown to negatively affect client activity[21]. Since loss of HCH1 relieves the growth phenotype caused by this mutation, we asked whether Hch1 may similarly impair client chaperoning by Hsp90. Activity of two Hsp90 clients, glucocorticoid receptor (GR) and the constitutively active yeast kinase Ste11ΔN, was measured using reporter gene constructs in the presence or absence of HCH1 in Hsp82 WT or W585T expressing yeast. Loss of HCH1 in combination with either Hsp82 WT or Hsp82-W585T led to increased GR and Ste11ΔN reporter activity (Fig. 2a,b). To determine whether reduced client protein expression could explain the loss of client activity in the presence of HCH1, we assessed the steady-state level of client proteins in hsc82hsp82 and hch1hsc82hsp82 cells expressing either Hsp82 WT or W585T. We found that protein expression of both GR and Ste11ΔN correlated to some extent, although not perfectly, with their level of activity (Fig. 2a,b).

Expression of the constitutively active viral oncogenic kinase v-SRC in yeast is lethal and relies on functional Hsp90 (ref. 26). Therefore, we tested the effect of HCH1 deletion on GAL-induced v-SRC function in cells expressing either Hsp82 WT or Hsp82-W585T. As seen in Fig. 2c, and as previously reported[21], Hsp82-W585T mutation alone led to loss of v-SRC activity, as the yeast remained viable in the presence of galactose. However, loss of HCH1 in combination with Hsp82-W585T rescued the v-SRC-mediated growth defect. v-SRC accumulation and phosphorylating activity were also dramatically increased in the hch1hsp82-W585T strain (Fig. 2d).

### Hsp90-Y627 phosphorylation disrupts client interactions.
Hch1 is not conserved from yeast to humans, yet appears to play a significant role in client remodelling in yeast. Thus, we investigated a possible functional replacement mechanism for Hch1 in higher eukaryotes. In human cells, phosphorylation of Hsp90α-Y627 disrupts some client interactions[20]. Hsp90α-Y627 is phosphorylated by the Yes kinase. Profiling of the evolutionary appearance of YES1 and loss of HCH1 using the NCBI database's SmartBLAST tool indicated that both events occurred at the same approximate point in eukaryotic evolution (for example, acquisition of multicellularity, see Supplementary Fig. 2). To investigate the impact of human Hsp90α-Y627 phosphorylation on GR, Raf-1 and v-SRC interaction, 293A cells were transfected with Flag-Hsp90α WT, the Y627 non-phosphorylatable mutant Flag-Hsp90α-Y627F, or the phosphomimetic mutant Flag-Hsp90α-Y627E. Transfected cells were collected, lysed and Flag-complexes were isolated using anti-Flag antibody coupled to agarose and analysed using the indicated antibodies. An additional set of the same transfections included a construct expressing v-SRC. These cells were used in a pulldown experiment to assess the impact of Hsp90α-Y627 mutation on Hsp90 interaction with v-SRC. Hsp90α-Y627F had very little or no impact on client interactions compared to WT. In contrast, Hsp90α-Y627E displayed reduced kinase interactions (Fig. 3a,b). GR interaction with human Hsp90 was not impacted by either mutation.

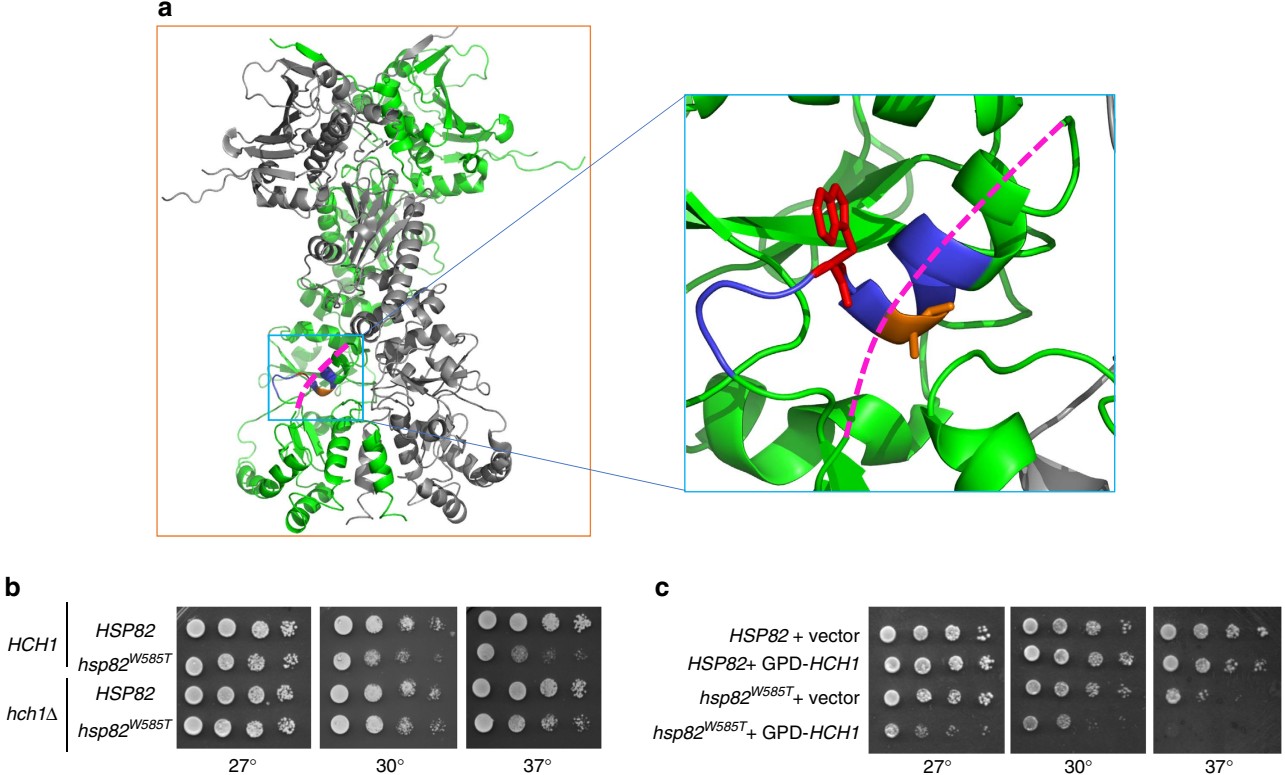

**Figure 1 | Loss of *HCH1* relieves the Hsp82-W585T growth phenotype.** See also Supplementary Fig. 1. (**a**) In the image on the left, one protomer of yeast Hsp82 is in green, the other in grey (PDB 2CG9). Residues 582QFGWSANME590 are in blue, with W585 in red and A587 in orange. The missing loop (which is not resolved in this crystal structure) containing Y606—598LRDSSMSSYMSSK610—is shown as a dotted line in magenta. The image on the right is a magnification of the middle domain/C-terminal domain interface of the Hsp82 protomer shown in the box in the left panel. (**b**) *hsc82hsp82* and *hch1hsc82hsp82* cells expressing a WT copy of *HSC82* were transformed with either WT or W585T *HSP82* constructs. Transformants were struck on to 5-FOA for two days, grown overnight in selective media, diluted 10-fold and grown on selective media plates at the indicated temperatures for two days. (**c**) *hsc82hsp82* cells expressing *HSP82* WT or W585T were transformed with vector or pRS416GPD-*HCH1*. Resulting colonies were grown as described in **b**.

**Hsp82-Y606E mutation complements Hch1 *in vivo* activity.** Hsp90-Y627 is conserved (Hsp82-Y606) but not phosphorylated in yeast[22]. Hch1's effect on client activity in yeast demonstrates functional similarities with Hsp90-Y627 phosphomimetic mutation in human cells, suggesting a possible evolutionary basis for the loss of Hch1 in higher eukaryotes. To explore this possibility further, we generated phosphomimetic (E) and non-phosphorylatable (F) mutations at position Y606 in plasmids containing *HSP82* WT or W585T. These plasmids were then expressed as the only form of Hsp90 in *HCH1* or *hch1Δ* yeast cells. Of note, the combination of Hsp82-Y606E and W585T was non-viable (Supplementary Fig. 3A). However, analysis using serial dilution growth assays demonstrated that the Hsp82-Y606E mutation alone resulted in a slight growth phenotype at 27° and 30° that was rescued by loss of endogenous *HCH1* (Supplementary Fig. 3B). In contrast, expression of Hsp82-Y606F mutation alone did not cause a growth defect. Next, we asked whether increased levels of Hch1 would exacerbate the Hsp82-Y606E growth phenotype. A plasmid expressing *HCH1* from the constitutive GPD promoter was transformed into the Hsp90 mutant strains containing endogenous *HCH1*. Figure 4a shows that increased *HCH1* expression had no effect on cells expressing either Hsp82 WT or Y606F, but did have a clear effect on cells expressing Hsp82-Y606E. Taken together, these data are consistent with our hypothesis that phosphorylation of this amino acid in higher eukaryotes may play a role similar to that of Hch1.

To further explore this possibility, we asked whether Hsp82-Y606E could rescue a mutation that is relieved by Hch1. Hch1 has been identified as a suppressor of the Hsp82-E381K mutation[27]. Thus, we introduced the E381K mutation into *HSP82* WT, Y606F or Y606E constructs, which we then expressed in *hsc82hsp82* cells. We compared cell growth by serial dilution of overnight cultures followed by growth on either rich media or 5-FOA (to lose the WT copy of Hsp90). Although the Hsp82-E381K mutant and Hsp82-E381K-Y606F mutant displayed similar growth deficiencies on the 5-FOA plate, introduction of Y606E partially relieved the growth defect caused by E381K mutation (Fig. 4b).

Next, we compared the ability of *HCH1* expression and introduction of Y606E mutation to rescue the growth phenotype associated with Hsp82-E381K. *hsc82hsp82* cells expressing Hsp82 WT, E381K or E381K-Y606E as their sole source of Hsp90 were transformed with vector or GPD-*HCH1*. Transformed cells were then grown at 27°, 30° or 37° as indicated. Figure 4c demonstrates that GPD-*HCH1* is able to fully rescue the growth phenotype of the Hsp82-E381K mutant at 27° and 30°, and partially rescue growth at 37°. The addition of Y606E to Hsp82-E381K was likewise able to relieve growth at 27° and 30°, although not to the extent of GPD-*HCH1*, and was not capable of rescuing growth at 37°. The ability of the Hsp82-Y606E mutation to partially relieve the E381K-mediated growth defect is consistent with the possibility that this phosphomimetic mutation in *Saccharomyces cerevisiae* shares some functional similarity to Hch1.

**Hsp82-Y606E disrupts client protein expression and activity.** We next explored whether phosphomimetic mutation of Hsp82-

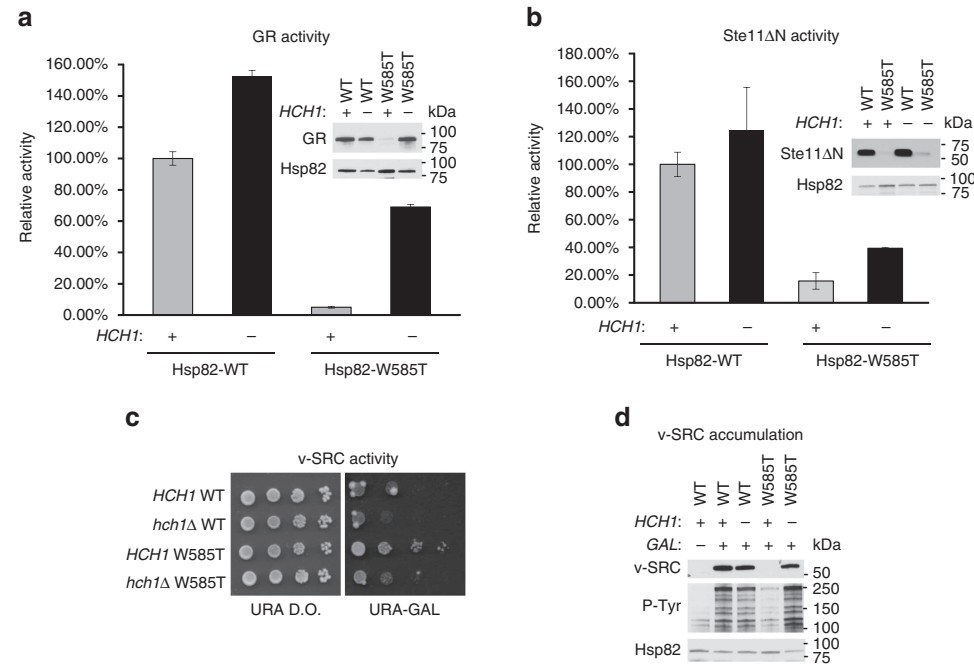

**Figure 2 | Hch1 reduces client accumulation and activity.** (**a**) *HCH1* (MR318) or *hch1* (CLY7) cells expressing Hsp82 WT or W585T (MR923-MR926) were transformed with a plasmid encoding the mammalian steroid hormone receptor GR and the β-galactosidase reporter pUCΔSS-26X. Resultant colonies were grown to mid-log phase followed by the addition of synthetic hormone. Following a five-hour incubation, β-galactosidase activity was quantified (each sample was run in triplicate and error bars represent s.d. from the mean). These same strains were lysed and analysed using SDS–PAGE and the indicated antibodies. (**b**) The same strains used in **a** were transformed with the GAL-inducible Ste11ΔN construct as well as the β-galactosidase reporter PRE-lacZ (ref. 55). Transformants were grown to mid-log phase in selective media supplemented with raffinose followed by a 6-h galactose induction. Cells were then measured for β-galactosidase activity as in **a**. These strains were also lysed following galactose induction and run on a 4–20% SDS gel and probed with the indicated antibodies. (**c**) *HCH1* or *hch1* cells expressing either WT or W585T Hsp82 as indicated were transformed with pRS316GAL-v-SRC. Transformants were grown overnight in minimal media and then diluted 10-fold and grown for four days at 30° on minimal media supplemented with glucose (URA D.O.) or galactose (URA-GAL). (**d**) Cells from **c** were grown overnight in URA-RAF and switched to URA-GAL media overnight. Cells were lysed and analyzed as in **b**.

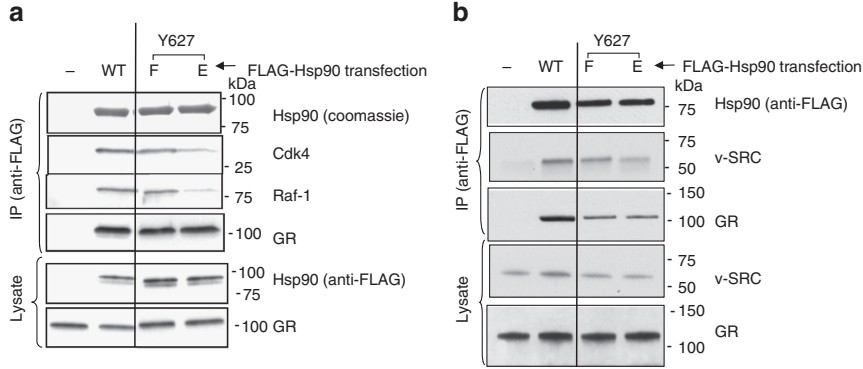

**Figure 3 | Hsp90-Y627 phosphorylation disrupts client interactions.** See also Supplementary Fig. 2. (**a**) 293 A cells were transfected with the pcDNA3-Flag vector or Flag-Hsp90α WT, Y627F or Y627E as indicated. Cells were collected, lysed and Flag complexes were isolated using anti-Flag resin and analysed using SDS–PAGE. (**b**) Transfections were performed and analysed as in **a**. Except a pLNCX-chick-v-SRC construct was co-transfected with each sample.

Y606 may phenocopy the impact of Hch1 on client remodelling in *S. cerevisiae*. GR and Ste11ΔN activity were measured as in Fig. 2. Hsp82-Y606 mutation markedly affected GR activity (Fig. 5a). While deletion of *HCH1* relieved the impact of Hsp82-Y606F on GR activity, it only minimally rescued GR activity in the *hsp82*-Y606E strain.

Ste11ΔN activity was not affected in the *hsp82*-Y606F strain (compared to Hsp82 WT). However, its activity was strongly reduced in the *hsp82*-Y606E strain (Fig. 5b). Consistent with the data in Fig. 5a, loss of *HCH1* only slightly rescued Ste11ΔN

activity in the presence of Hsp82-Y606E. As in Fig. 2, GR and Ste11ΔN protein expression correlated to some extent with their activity, although the correlation between GR activity and its accumulation was less pronounced (Fig. 5a,b). v-SRC activity was monitored using yeast growth assays and western blots to visualize its accumulation and kinase activity in *hsp82*-Y606F and *hsp82*-Y606E yeast strains. As demonstrated by cell growth in Fig. 5c, v-SRC is less active in *HCH1* cells expressing the Hsp82-Y606E mutant. Protein expression and phosphorylating activity of v-SRC were monitored in these strains as well. v-SRC protein

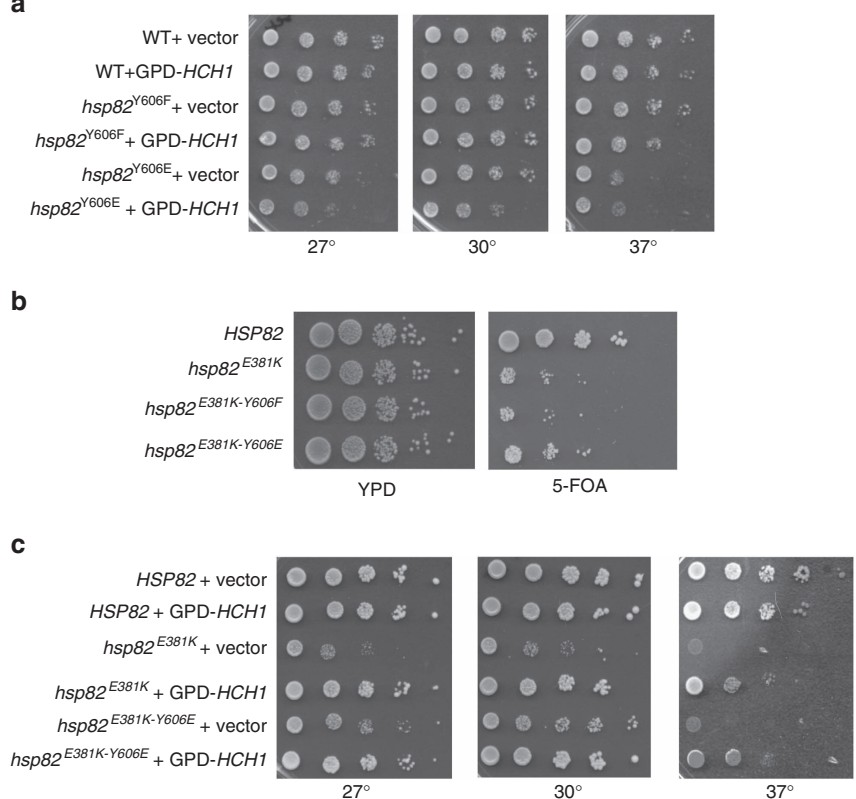

**Figure 4 | The Hsp82-Y606E mutation complements Hch1 *in vivo* activity.** See also Supplementary Fig. 3. (**a**) *hsc2hsp82* cells were transformed with Hsp82 WT, Y606F or Y606E constructs. The transformed cells were then grown on 5-FOA to lose the WT copy of Hsp90 in those strains and transformed with either vector or pRS416GPD-*HCH1*. The second set of transformants were then grown overnight in selective media, diluted 10-fold and grown for 2 days at the indicated temperatures. (**b**) *hsc2hsp82* cells were transformed with the indicated *HSP82* mutants and grown on 5-FOA for 2 days. Colonies grown on 5-FOA were then grown overnight in minimal media, diluted 10-fold and grown on YPD or 5-FOA plates for two days at 30°. (**c**) *hsc2hsp82* cells transformed with the indicated mutants as well as vector or WT pRS416GPD-*HCH1* were grown as in **a** on selective media.

and activity increased in both the *hsp82*-Y606F and *hsp82*-Y606E mutant strains in the absence of *HCH1*, although presence of Hsp82-Y606E lessened the impact of *HCH1* deletion, consistent with the data above (Fig. 5d).

Hch1 was shown previously to increase cell sensitivity to Hsp90 inhibitors, especially in the context of Hsp82-A587T mutation[23]. Therefore, we asked whether loss of *HCH1* would impact drug sensitivity of the *hsp82*-W585T, Y606F and Y606E mutant strains. *hsc2hsp82* or *hch1hsc2hsp82* cells expressing either Hsp82 WT, W585T, Y606F and Y606E mutants were diluted 10-fold onto plates containing either DMSO (0) or the indicated amount of the Hsp90 inhibitor radicicol. First, consistent with the earlier findings, we found that both Hsp82-W585T and Hsp82-Y606E mutant yeast displayed increased sensitivity to the Hsp90 inhibitor, while Hsp82-Y606F expressing yeast did not (Supplementary Fig. 4). Further, loss of *HCH1* partially relieved the drug sensitivity of the *hsp82*-W585T and Y606E mutant strains. Taken together, these findings suggest that the helix and loop region containing both W585 and A587 represents an important, Hch1-influenced determinant of Hsp90 inhibitor sensitivity in yeast.

**A107N mutant rescues Hsp82-W585T and *HCH1* phenotypes.** Our data are consistent with the possibility that Hch1 and Hsp82-W585T mutation both promote an alteration in Hsp90 conformation that negatively affects client remodelling while enhancing drug sensitivity. Previous data suggest that such a scenario is consistent with stabilization of the open Hsp90

conformation[28]. To provide experimental support for this possibility, we utilized a mutation which promotes the closed Hsp90 conformation, and we asked whether it could rescue the phenotypes we have described above. A107N is such a mutation, as it results in stabilized ATP lid closure and enhances Hsp90 N-domain dimerization, thus favouring the closed conformation[29–31]. We measured GR activity in yeast expressing Hsp82 WT, Hsp82-A107N, Hsp82-W585T or Hsp82-A107N-W585T (Fig. 6a). Further, each of these strains were either made deficient in *HCH1* (−), expressed endogenous *HCH1* (+), or overexpressed *HCH1* (arrow pointing up). In Hsp82 WT and mutant strains, overexpression of *HCH1* uniformly reduced GR activity. In contrast, *HCH1* deletion uniformly increased GR activity in all Hsp82 strains (consistent with the data in Fig. 2a). As we predicted, Hsp82-A107N mutation rescued GR activity in Hch1 overexpressing cells and the double mutant Hsp82-A107N-W585T restored GR activity to WT levels. Although loss of *HCH1* enhanced these effects, further stimulation of GR activity in the presence of Hsp82-A107N compared to Hsp82 WT was no longer evident.

In *S. cerevisiae*, cells expressing Hsp82-A107N are resistant to Hsp90 inhibitors[29]. Therefore, we asked whether introduction of A107N would reverse the enhanced drug sensitivity of *hsp82*-W585T yeast cells in the presence of endogenous Hch1. We transformed the *hsc2hsp82* strain with plasmids expressing either Hsp82 WT, A107N or A107N-W585T. Consistent with our previous data, we found that, despite the presence of endogenous Hch1, the A107N mutation rescued the increased drug sensitivity incurred by the W585T mutation (Fig. 6b). Importantly, the slow

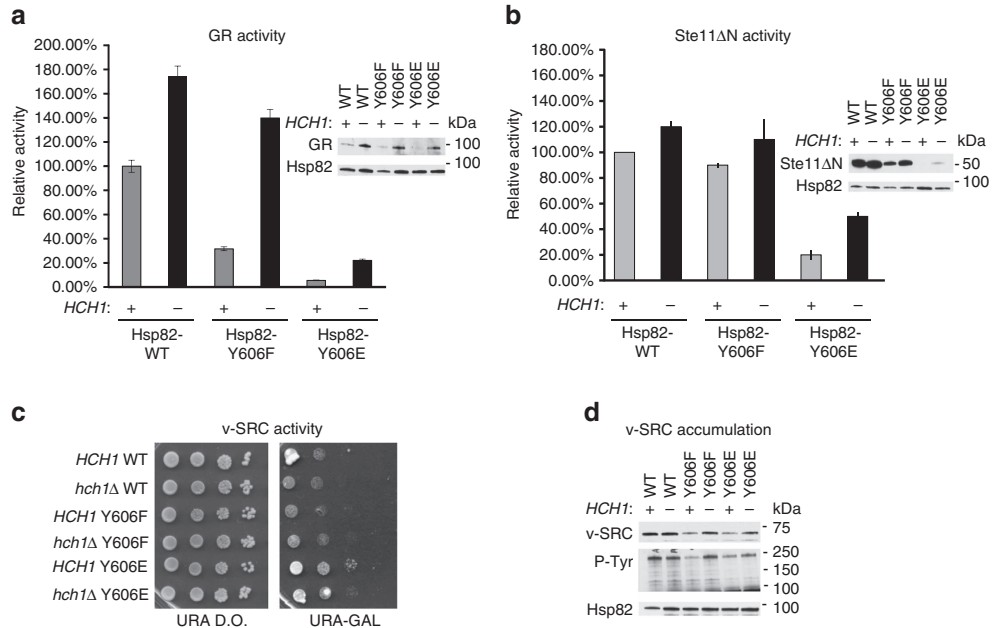

**Figure 5 | Hsp82-Y606E disrupts client protein expression and activity.** See also Supplementary Fig. 4. (**a,b**) hsc82hsp82 and hch1hsc82hsp82 cells expressing Hsp82 WT, Y606F or Y606E (MR923, MR925, CLY25, CLY29, CLY24 and CLY28 respectively) were transformed with plasmids containing either pG/N795-GR and pUCΔSS-26X (**a**) or pRS414GAL-Ste11ΔN and PRE-lacZ (**b**). Assays were performed in triplicate as indicated in the Fig. 2a,b. (**c**) HCH1 or hch1 cells expressing Hsp82 WT, Y606F or Y606E as indicated were transformed with pRS316GAL-v-SRC. Transformants were grown overnight in minimal media, diluted 10-fold and grown for 2 days at 30° on minimal media supplemented with glucose (URA D.O.) or galactose (URA-GAL). (**d**) Transformants from **c**. were grown overnight in Ura-RAF and switched to URA-GAL media overnight. Cells were lysed and analyzed as indicated.

growth phenotype characteristic of Hsp82-W585T (in the presence of DMSO) was also rescued by inclusion of the A107N mutation.

Since the Hsp82-A107N mutant prefers a closed conformation and appears to counteract the negative impact of Hch1 overexpression and Hsp82-W585T mutation on both client activity and drug sensitivity, these data support the possibility that Hch1 and Hsp82-W585T may function either in series or in parallel to stabilize the open conformation of Hsp90. To address this question, we expressed His-tagged Hch1 protein in yeast harbouring either Hsp82 WT or Hsp82-W585T, we pulled down Hch1 using nickel beads, and we blotted for Hch1-associated Hsp82. After normalizing for the amount of Hch1 pulled down, we found no clear difference in Hsp82 (Supplementary Fig. 5). These data are consistent with the likelihood that Hch1 and W585T mutation of Hsp82 exert similar but mechanistically distinct effects on chaperone function.

The nucleotide-dependent interaction of Hsp90 with its co-chaperone Sba1 (yeast p23) requires N-domain dimerization, and thus can be used to assess the closed/open status of Hsp90 in the presence of ATP[32,33]. We analysed His-Hsp82 WT and His-Hsp82-W585T interaction with Sba1 in yeast lysate in the presence of the non-hydrolyzable ATP analogue, AMP-PNP. As in Fig. 6a, this assay was performed in strains either deficient in HCH1(−), expressing endogenous HCH1 (+), or overexpressing HCH1 (arrow pointing up). Sba1 interaction with the Hsp82-W585T mutant was completely disrupted in the presence of endogenous Hch1, supporting the likelihood that this mutation disfavours N-domain dimerization. The impact of W585T mutation was only slightly relieved by deletion of HCH1. Although deletion of endogenous HCH1 did not affect Sba1 interaction with Hsp82 WT, overexpression of Hch1 markedly reduced this interaction (Fig. 6c). These data suggest that both the Hsp82-W585T mutation and overexpression of Hch1 (in the

context of Hsp82 WT) stabilize an open Hsp90 conformation that is deficient in client remodelling and displays increased drug sensitivity[28].

To determine if introduction of A107N mutation is able to reverse the conformational defects observed with Hsp82-W585T, we monitored Sba1 interaction in the presence and absence of nucleotide in cells expressing Hsp82 WT, A107N, W585T and A107N-W585T. Figure 6d demonstrates that, in comparison to WT cells, Hsp82-A107N mutation dramatically increases Sba1/Hsp82 interaction in the presence of nucleotide, consistent with published data showing that this mutation stabilizes the closed conformation. In the absence of nucleotide, A107N-W585T double mutation appears to favour Hsp82 occupancy of closed state 1 (characterized by increased association of Aha1 but not Sba1). Addition of nucleotide resulted in a full recovery of wild-type levels of Sba1 interaction with the Hsp82 double mutant, characteristic of the Hsp82 closed state 2 (ref. 16). Taken together, these data suggest that its ability to attain closed state 2 rescues the chaperone activity and slow growth phenotype characteristic of Hsp82-W585T.

**A107N restores function and conformation of Hsp82-Y606E.** To asses the ability of the A107N mutation to rescue the chaperone activity of the Hsp82-Y606E mutant, we examined GR activity in cells expressing Hsp82 WT, Y606E, A107N and A107N-Y606E. Although the A107N mutation was able to partially relieve the chaperoning defect of the Y606E mutation, full GR activity was not restored (Fig. 7a). This is not surprising as the Y606E mutation has a more severe impact on client activity and growth compared to overexpression of HCH1.

Finally, we examined nucleotide-dependent Sba1 interaction with Hsp82-Y606F and Hsp82-Y606E mutants in yeast expressing endogenous Hch1. While Hsp82 WT and Y606F bound

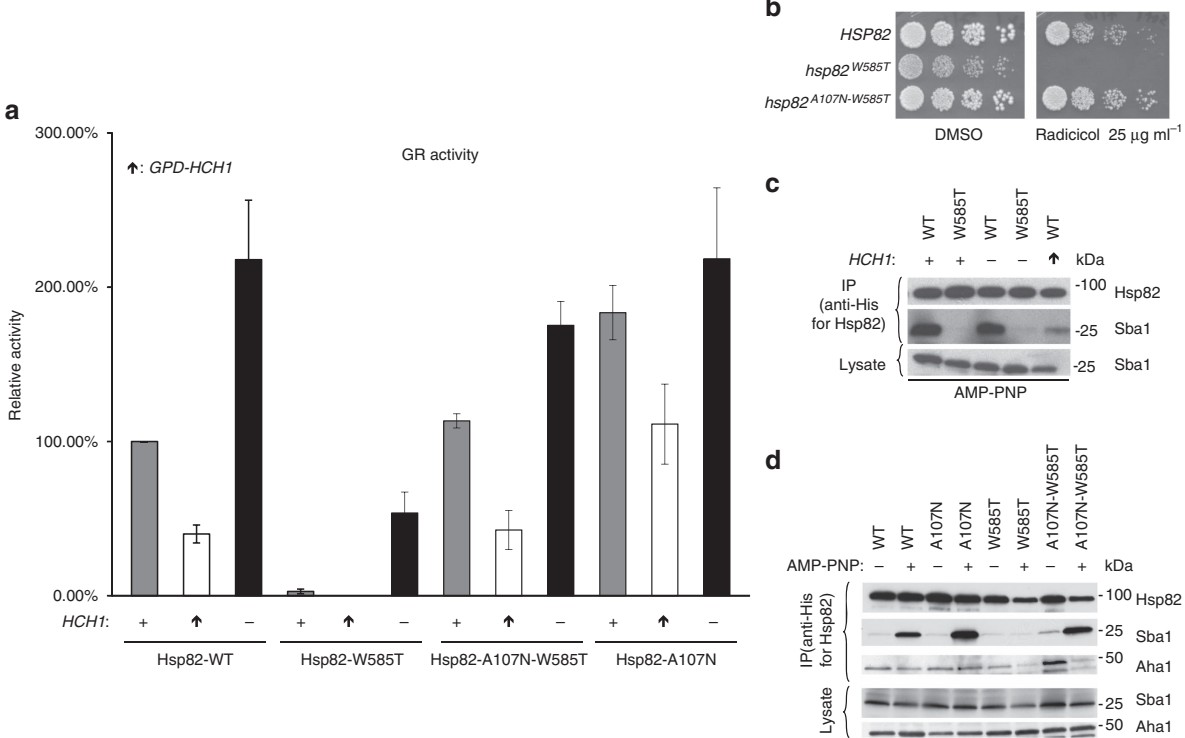

**Figure 6 | A107N mutant rescues Hsp82-W585T and *HCH1* phenotypes.** See also Supplementary Fig. 5. (**a**) *hsc82hsp82* and *hch1hsc82hsp82* cells expressing Hsp82 WT (MR923 and MR925), W585T (MR851 and MR858), A107N-W585T (MR852 and MR859) or A107N (AF3 and AF6) were transformed with vector or pRS414GPD-*HCH1* (arrow pointing up) as well as a plasmids containing pG/N795-GR and pUCΔSS-26X. These cells were grown to mid-log phase followed by the addition of synthetic hormone. Following a 5-h incubation, β-galactosidase activity was quantified (each sample was run in triplicate and error bars represent s.d. from the mean). These strains were then lysed and analysed using SDS–PAGE and the indicated antibodies. (**b**) *hsc82hsp82* (MR318) cells were transformed with either *HSP82* WT, W585T or A107N-W585T constructs. Transformants were struck on to 5-FOA for 2 days, grown overnight in rich media, diluted 10-fold and grown on rich media plates containing DMSO or 25 μg ml$^{-1}$ of the Hsp90 inhibitor Radicicol and incubated at 30° for 2 days. (**c,d**) *hsc82hsp82* and *hch1hsc82hsp82* cells were transformed with the indicated His-tagged Hsp82 constructs as well as vector or pRS414GPD-*HCH1* (arrow pointing up). Cells were grown overnight in minimal media and collected. These cells were then lysed and His-complexes were isolated using nickel resin beads in the presence (+) or absence (−) of AMP-PNP. Following SDS–PAGE, His-complexes were analyzed by Western blot with antibodies shown.

comparable levels of Sba1 in the presence of AMP-PNP, Hsp82-Y606E/Sba1 interaction was dramatically reduced, equivalent to the impact of Hch1 overexpression (Fig. 7b). Consistent with these results and supporting a model in which phosphomimetic mutation of human Hsp90α in its C-terminal domain stabilizes an open, N-domain undimerized conformation, we found that, in co-immunoprecipitations from human cell lysate, Hsp90α-Y627E bound markedly less p23 than did either Hsp90α-Y627F or WT Hsp90 (Supplementary Fig. 6). Analogous to the data in Fig. 6d, addition of the A107N mutation to Hsp82-Y606E resulted in markedly increased Sba1 interaction in the presence of nucleotide (Fig. 7c). After normalizing to the amount of Hsp82 pulled down, the amount of Sba1 interacting with Y606E is 36.5% of that bound to WT Hsp82, while the amount of Sba1 co-precipitating with Hsp82-A107N-Y606E is increased by 6.6-fold compared to the amount bound to Hsp82-Y606E. These data further support the hypothesis that Hch1 and Y606E negatively impact the Sba1-stabilized closed conformation of Hsp90 and reversal of this defect results in improved Hsp90 function.

## Discussion

*S. cerevisiae* has been a valuable genetic tool to better understand the impact of Hsp90 mutation or the loss/overexpression of co-chaperone proteins on cellular fitness. Synthetic growth phenotypes and Hsp90-client activity assays can be used to

detect redundant or opposing cellular functions within this chaperone network. In this study, we have used this approach to provide evidence that the co-chaperone *HCH1* exacerbates the growth phenotype of the client-remodelling mutant Hsp82-W585T, and that both the co-chaperone and this Hsp82 mutation exert similar long-range effects on Hsp90 conformation.

Amino acids 582–590 in Hsp82 are highly conserved and comprise a part of a hydrophobic, solvent exposed, flexible helix and loop at the interface between the middle and C-terminal domains that appears to participate in substrate binding in *E. coli* Hsp90 (HtpG)[21,34]. In Hsp82, the hydrophobic character of this region is important since polar mutations at positions 583 (phenylalanine), 585 (tryptophan) or 587 (alanine) negatively affect yeast growth[23,35]. Although mutations within this region (for example, W585T) clearly impact Hsp90-dependent client function, their direct effect on binding of an artificial client is minimal[21], suggesting an alternative mechanistic basis for their phenotypes. Based on reduced ATPase activity of these Hsp82 mutants, Genest and colleagues speculated that mutations within this region may have long-range conformational effects on this structurally dynamic chaperone. Our data demonstrating (1) markedly reduced binding of Sba1 (requires dimerized Hsp90 N-domains for binding) to Hsp82-W585T, and (2) restoration of GR activity to levels supported by Hsp82 WT on introduction of an A107N mutation (favouring ATP lid closure and subsequent N-domain dimerization) into Hsp82-W585T, together with

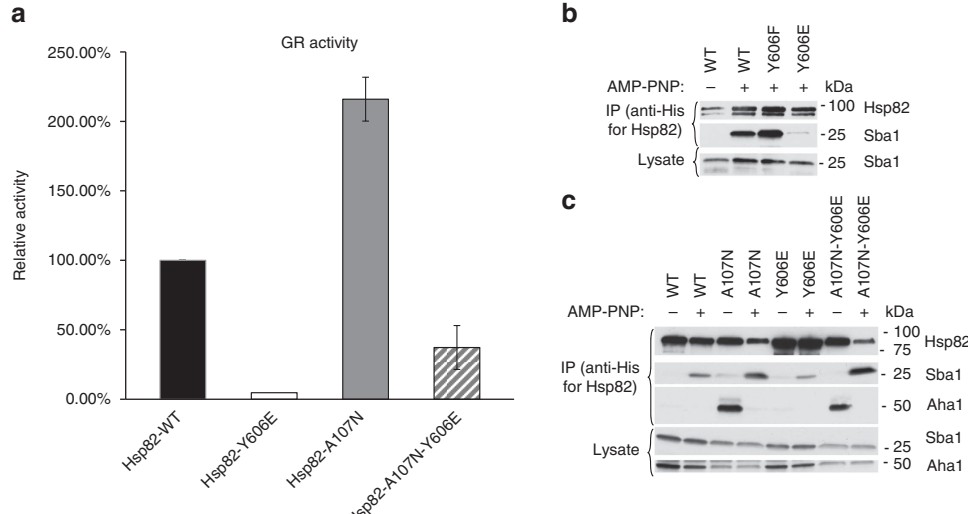

**Figure 7 | A107N restores function and conformation of Hsp82-Y606E.** See also Supplementary Fig. 6. (**a**) *hsc82hsp82* cells expressing Hsp82 WT (MR923), Y606E (CLY24), A107N (AF3) and A107N-Y606E (MR1016) were transformed pG/N795-GR and pUCΔSS-26X. Resultant colonies were grown to mid-log phase followed by the addition of synthetic hormone. Following a 5-h incubation, β-galactosidase activity was quantified (each sample was run in triplicate and error bars represent s.d. from the mean). These same strains were lysed and analysed using SDS–PAGE and the indicated antibodies. (**b,c**) *hsc82hsp82* cells were transformed with the indicated His-tagged Hsp82 constructs. Resultant colonies were grown overnight in rich media and cell pellets were collected. Collected cell pellets were lysed and His-complexes were isolated using nickel resin in the presence (+) or absence (−) of AMP-PNP. Following SDS–PAGE, His-complexes were analysed by Western blot as in Fig. 6.

restoration of Sba1 binding to wild-type levels in the double mutant, support this possibility and support a model in which W585T mutation induces Hsp82 to preferentially occupy an open conformation, negatively affecting its ability to engage in a productive chaperone cycle.

Hsp90 conformational dynamics and chaperoning are highly dependent on co-chaperone assistance. Thus, Cdc37 and Hop (ySti1) bind to the open conformation of Hsp90 and play roles in client loading, while p23 (ySba1) binds to the closed conformation and increases the Hsp90-client interaction time[6–10,17–19]. Likewise, Aha1 binding promotes ATP hydrolysis by facilitating the initial steps in the conformational rearrangement of ATP-bound Hsp90 (refs 13,36,37). Loss of any of these co-chaperone functions leads to reduced client protein remodelling and activity[38–44].

The yeast co-chaperone Hch1, while somewhat homologous to Aha1 in sequence and binding to the same site in the middle domain of Hsp90, lacks Aha1's second binding site in the Hsp90 N-domain and is a poor activator of Hsp90 ATPase[36]. Although an earlier study showed that *HCH1* could rescue the growth defect caused by E381K mutation[27], and a more recent paper suggested a role for Hch1 in modulating yeast sensitivity to Hsp90 inhibitors[23], the lack of *HCH1* expression in metazoans has led investigators to question its importance for Hsp90 function.

ATP-initiated Hsp90 domain rearrangements and productive Hsp90 conformational cycling require repositioning of middle domain contacts with both the N and C-terminal domains[45–47]. Our data suggest that Hch1 may play a role in regulating Hsp90 conformational dynamics by stabilizing the open conformation, perhaps by restricting the mobility of the Hsp90 middle domain. W585T mutation may generate a similar phenotype by disrupting middle and C-terminal domain interactions. Given this scenario, it is possible to envision how Hch1 might exacerbate the growth and client defects of Hsp82-W585T mutation. Further, this model demonstrates how *HCH1* overexpression in the background of Hsp82 WT might reproduce some of the client and conformational phenotypes caused by W585T mutation.

Several PTM sites in human Hsp90 are conserved in *S. cerevisiae*, although they are not modified in yeast[22,48]. This may reflect the increased complexity of Hsp90 regulation and function during eukaryotic evolution. We previously reported that client dissociation from Hsp90 appears to be a consequence of phosphorylation of Y627 in the human chaperone[20]. This residue is located in a highly flexible loop region within the Hsp90 C-terminal domain and, although no crystal structure containing this region is available, recent cryo-electron microscopy data suggest that this loop lies in close proximity to the hydrophobic loop and helix region of which W585 is a part (W595 in human Hsp90α; see Fig. 1a)[49]. Using the NCBI database, we found that the appearance in eukaryotic evolution of the kinase (c-Yes), which we previously identified as able to phosphorylate Hsp90-Y627 (ref. 20), coincides with both the loss of *HCH1* and the appearance of multicellularity. Consistent with this observation, although the equivalent tyrosine in yeast Hsp82 (Y606) is not phosphorylated[22], our data demonstrate that phosphomimetic (but not non-phosphorylatable) mutation at this position (Y606E) causes similar growth and client defects as are observed with *HCH1* expression. Not surprisingly, based on the model presented in Fig. 1a, introduction of Y606E into Hsp82-W585T is lethal. However, deletion of *HCH1* partially rescues growth, client and drug phenotypes caused by Hsp82-Y606E. Further, like Hch1, Hsp82-Y606E partially rescues the growth defect caused by Hsp82-E381K. Of note, while several kinase clients fail to interact with Hsp90-Y627E in human cells, interaction with GR is unimpeded. This is in distinct contrast to the deleterious impact of yeast Hsp82-Y606E mutation on GR activity and protein expression, and supports the concept of greater divergence in client chaperone requirements in human compared to yeast cells. We also demonstrate that, like Hch1, Hsp82-Y606E mutation similarly stabilizes an open yeast Hsp90 conformation (as evidenced by reduced Sba1 binding), which is consistent with increased sensitivity to Hsp90 inhibitors and defective client processing in yeast expressing this mutant as the sole Hsp90 protein. In agreement with this model, Y627E mutation of human Hsp90α abrogates interaction with both

Aha1 (ref. 20) and p23 (this study). In sum, our data support a role for Hch1 in modulating yeast Hsp90 conformational dynamics. Further, we propose that the flexible helix and loop region adjacent to the C-terminal Hsp90α dimerization domain is a key modulator of N-domain rearrangements and that phosphorylation-mediated reversible conformational modulation of this region provides an evolutionary replacement for Hch1 in metazoans.

## Methods

**Media composition as well as chemicals and antibodies.** Yeast was grown in either YPD (1% Bacto yeast extract, 2% peptone, 2% dextrose) or defined synthetic growth media supplemented with 2% dextrose, galactose or raffinose where indicated[50]. Standard genetic methods were applied. Yeast growth assays were performed by growing cells overnight at 30° followed by serial 10-fold dilutions on to the appropriate media and growing at the indicated temperatures for 2 days, unless otherwise specified. 5-FOA was obtained by Zymo Research Chemicals. Radicicol was purchased from Sigma. The Sba1 and Aha1 antibodies were previously published[32,51]. The Hsp82 antibody was purchased from Stress Marq Biosciences (clone Hyb-K41220A). The Hsp82 antibody used in Fig. 6c was obtained by (Enzo ADI-SPA-840). The v-SRC antibody was purchased from Upstate Scientific (clone EC10). The phospho-tyrosine antibody was acquired by Millipore (clone 4G10). The Cdk4 and Raf-1 antibodies were purchased from Santa Cruz Biotechnology. The BuGR2 (α-GR) antibody was purchased from abcam. The Xpress and Flag antibodies were purchased from Thermo Fisher/Pierce. The p23 antibody was purchased from Affinity Bioreagents. All antibodies were diluted 1:1,000, uncropped scans of each Figure can be found in Supplementary Fig. 7.

**Yeast strains.** All strains are isogenic to WT W303 or 779-6A (Ste11ΔN, GR and drug growth assays). ΔPCLD82a (MATa ade2-1 leu2-3,112 his3-11,15 trp1-1 ura3-1 can1-100 hsc82::LEU2 hsp82::LEU2), ΔPCLD82aΔhch1 (MATa ade2-1 leu2-3,112 his3-11,15 trp1-1 ura3-1 can1-100 hch1::HIS3 hsc82::LEU2 hsp82::LEU2) and MR318 (MATα hsp82Δ::KanMX4 hsc82Δ::KanMX4 + pRS416HSP82p-HSP82) have been previously reported[21,23,26]. ΔPCLD82a and ΔPCLD82aΔhch1 were used to examine growth and protein interaction. MR318 was used to generate CLY7 (MATα hsp82Δ/hsc82Δ::KanMX4 hch1::: His3MX6 + pRS416HSP82p-HSP82). MR318 and CLY7 were used to further generate the following strains to observe Ste11ΔN, GR, and drug growth assays: MR923, MR924, MR925, MR926, CLY24, CLY25, CLY28, CLY29, MR851, MR852, MR858, MR859, AF3, AF6 and MR1016. A detailed list of the strains and plasmids used can be viewed in Supplementary Tables 1 and 2 of the Supplementary Experimental Procedures. Plasmids containing mutated genes were generated using QuickChangeII site directed mutagenesis (Stratagene). Primers sequences are listed in Supplementary Table 3. A quick reference guide to the Hsp82 mutations used in this study and their functional properties can be found in Supplementary Table 4.

**Yeast growth.** In *S. cerevisiae*, two genes, *HSC82* and *HSP82*, encode Hsp90. Deletion of both genes results in cell lethality. To introduce Hsp90 mutations, *hsc82hsp82* and *hch1hsc82hsp82* cells containing a rescue *HSC82*-URA3 plasmid were transformed with *HSP82* WT or mutant constructs and the rescue plasmid was selected against using 5-FOA. In the Radicicol growth assay, Radicicol was dissolved in DMSO and added to YPD + agar media for the final concentrations of 25 µg ml$^{-1}$. Overnight cultures of cells were diluted to OD600 = 0.5 followed by 10-fold serial dilutions and spotting onto YPD + Radicicol and YPD + DMSO plates and put at 30° for 2–3 days.

**β-Galactosidase activity assays.** CLY7 and MR318 strains carrying the indicated *HSP82* alleles were transformed with plasmids pG/N795 encoding mammalian steroid receptor GR and the β-galactosidase reporter pUCΔSS-26X (ref. 52). Overnight cultures of three independent transformants from each strain were diluted to 0.4 OD$_{600}$ in fresh media and synthetic hormone analogue deoxycorticosterone (Sigma) was added to the final concentration of 10 µM. After 5-hours of incubation, the β-galactosidase assay was performed and corrected by cell density[53,54]. Cells from 1 ml of culture were harvested by centrifugation, washed with water and resuspended in 150 µl of Z-buffer (60 mM Na$_2$HPO$_4$, 60 mM NaH$_2$PO$_4$, 10 mM KCl, 1 mM MgSO$_4$, 50 mM β-ME). Cells were then permeabilized by the addition of 50 µl chloroform and 20 µl of 0.1% SDS followed by vigorous vortexing for 30 s. 700 µl of ONPG (2-Nitrophenyl β-D-galactopyranoside, Sigma; 1 mg per ml in Z-buffer) was added and incubated at 30° for 10 min. The reaction was stopped by adding 500 µl of 1 M Na$_2$CO$_3$ and the absorbance was measured 420 nm. Mixed solution containing all reagents without cells served as the blank, and processed cells without hormone treatment as the internal control. β-galactosidase activity is calculated in Miller units by the equation: {1,000 × A$_{420}$ per [OD$_{600}$ × culture volume (ml) × reaction time ($_{min}$)]} relative to wild type. All assays were performed in triplicate.

**Ste11 activity.** *HCH1* or *hch1* yeast strains containing the indicated *HSP82* alleles were transformed with pRS414GAL-Ste11ΔN and the β-galactosidase reporter PRE-lacZ (ref. 55). Overnight cultures of three independent transformants were grown in raffinose selective media and diluted to 0.4 OD600 in fresh medium. Ste11ΔN was induced by adding galactose (2% for final concentration). After 5-h of induction, cells from 1 ml of culture were collected by centrifugation, washed with water and the β-galactosidase activity was measured as described for the GR activity assay.

**Galactose inductions for collecting lysates.** Yeast cells expressing pRS316GAL-v-SRC and pRS414GAL-Ste11ΔN were grown overnight in defined minimal growth media supplemented with raffinose to mid-log phase. On reaching mid-log phase, the media was exchanged for defined minimal media containing galactose overnight. Following the overnight induction, cells were collected and lysed for protein analysis using sodium dodecyl sulfate polyacrylamide gel electrophoresis (SDS–PAGE). Full immunoblot images can be referred to in Supplementary Fig. 7.

**Flag and nickel resin pulldown experiments A.** Flag-complex isolation: 293A cells (Invitrogen) cultured in DMEM were transfected at 80–90% confluency using 2 µg of the indicated plasmids using Lipofectamine 3000 Transfection Reagent (Life Technologies) overnight. Cells were collected and lysed using 50 mM Tris pH 7.5, 0.1% NP40, 1 mM MgCl$_2$, 100 mM NaCl plus protease inhibitor cocktail (Roche) and phosSTOP (Roche) followed by incubation with M2 anti-Flag antibody resin (Sigma-Aldrich) at 4° for 2 h to isolate the Flag-complexes. Flag-bound complexes were again washed with 50 mM Tris pH 7.5, 0.1% NP40, 1 mM MgCl$_2$, 100 mM NaCl plus protease inhibitor cocktail (Roche) and phosSTOP (Roche) to reduce background binding to the resin. Flag-protein complexes were than analysed by running a 4–20% SDS–PAGE gel, transferring on to a nitrocellulose membrane followed by immunoblotting with the indicated antibodies. B. His-complex isolation: The *hsc82hsp82* or *hch1hsc82hsp82* strains were transformed with His-Hsp82 constructs as shown in Figs 6 and 7. Transformants were struck onto 5-FOA to lose the WT copy of Hsp82 in those strains. Cells were taken from the 5-FOA plate and grown overnight in minimal media to OD$_{600}$ of 1.2–2.0 and lysed in 20 mM Tris (pH 7.5), 100 mM KCl, and 5 mM MgCl$_2$ plus protease inhibitor cocktail (Roche) and phosSTOP (Roche). Protein levels were adjusted to contain 5 mM AMP-PNP (Sigma-Aldrich) and incubated at 30° for 5 min. His-complexes were than extracted by incubating the lysate with nickel resin (Clontech) for 2 h at 4°. Following incubation, the resin was next washed with lysis buffer plus 0.1% Tween-20 and 35 mM imidazole. Protein complexes were analysed as in A. Full immunoblot images can be referred to in Supplementary Fig. 7.

**Data availability.** UniProt accession codes P02829, P53834, P04150, P23561, P00524, P07900 and PDB accession code 2CG9 were used in this study. All other data are available from the corresponding author upon reasonable request.

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

## Acknowledgements

This research was supported by the Intramural Research Programs of the National Cancer Institute and the National Institute of Diabetes and Digestive and Kidney Diseases.

## Author contributions

A.D.Z., M.R., C.L., W.X., S.A., D.J.L. and G.M.R.-M. conducted experiments for the paper. A.D.Z., M.R., C.L. and K.B. performed supplemental data experiments. A.D.Z., M.R., C.L., T.P., J.B.T, S.L., P.L. and J.J. created strains and plasmids used for the experiments. D.M. and L.N. assisted in designing and mentoring the project. A.D.Z. and L.N. wrote the paper.

## Additional information

**Competing interests:** The authors declare no competing financial interests.

