## [Peer review file · Nature Communications]

Reviewers' comments:

Reviewer #1 (Remarks to the Author):

In this paper, Zuelke et al. examine the role of tyrosine 627 phosphorylation on Hsp90 in mammalian and yeast cells. The authors conclude that phosphorylation of this site alters client and co-chaperone binding through a long-range conformational change that inhibits N-terminal dimerization. Fascinatingly their data suggest that lower eukaryotes bypass the need for Y627 phosphorylation through the presence of HCH1, a yeast-specific Hsp90 co-chaperone. While the concepts presented in this manuscript are exciting (clarifying the role of chaperone PTMs and the idea that they can functionally be replaced by co-chaperones), the authors do not go far enough to conceptually advance the field. As such, this manuscript is not suitable for publishing in Nature Communications in its current form. Specific comments:

- There are two main isoforms of Hsp90 in human, alpha and beta. The authors only examine Hsp90 alpha, which they label Hsp90 in this study. The manuscript should be adjusted to clarify this.
- Given studies showing differential binding of clients for Hsp90 isoforms, it would be interesting to see the effect of mutating the equivalent residue in Hsp90 beta (Y619). It would be simple to repeat the experiment in Figure 3 using tagged Hsp90 beta.
- The impact of Hch1 on the W585T has already been examined in Armstrong et al., 2012. The data in figure 1 do not show anything new about this regulation and should be at the very least placed into supplemental data, if not entirely removed.
- The altered interactions of Cdk4 and Raf-1 have already been published by Xu et al., 2013 and add nothing new to the mechanism of action.
- There is a discrepancy in the blots shown in Figure 3a. In the control lane in 3a, there is a GR band missing in the lysate blots. In 3b, showing a similar experiment, there is a GR band in the control lysate lane. How do the authors explain this?
- The authors demonstrate altered binding of Cdk4/Raf-1, but not GR. With only a few examples, it is hard to assess the impact of Y627 phosphorylation on client binding. The authors need to show present more examples using IPs of Hsp90 against defined Hsp90 interactors such as Ulk1, ER and c-Src.
 - Global proteomics studies of Hch1 interactors (similar to studies as Echtenkamp et al., 2011 and Sun et al., 2015) would provide additional insight into the role of Hch1 in the cell and substrate specificity.
- Given the conclusion of the paper that the need for Y627 phosphorylation in yeast is bypassed via Hch1, it would be interesting to examine the effects of expressing yeast Hch1 in mammalian cells. Expression of Hch1 in mammalian cells should phenocopy Y627E (i.e. altered co-chaperone and client binding)
- In 6B, W585T cells demonstrate sensitivity to Radicicol. Does deletion of Hch1 restore growth?
- The authors use genetic analysis to suggest Y627 phosphorylation alters Hsp90 dimerization. Hsp90 dimerization should be examined through more direct biochemical methods including co-IP of differentially tagged monomers as in Mollapour et al. 2015.

Reviewer #2 (Remarks to the Author):

In this study the authors show that the Hsp90 W585T mutant disrupts client remodeling and stabilizes Hsp90 in an open conformation. They suggest that the yeast co-chaperone Hch1 stabilizes the open conformation and thereby exacerbates growth, drug sensitivity and client activation of the Hsp90 W585T mutant. Finally, they show that phosphorylation of Y627 in human Hsp90 (Y606 in yeast) has a similar effect as Hch1 and they suggest that this PTM has a function in metazoans similar to that of Hch1 in yeast.

This is an interesting concept and the experiments are well performed.

Specific comments

1. Hch1 was previously described as a weak stimulator of Hsp90's ATPase activity – this seems to be in conflict with the authors' model, that Hch1 keeps Hsp90 in an open conformation. Direct evidence of the open state in the presence of Hch1 would be interesting.

2. The authors mention in the text that the GR and Ste11 Δ N expression level corresponded to the level of activity. In figure 2 this is only true for Ste11 Δ N since GR levels in the W585T hch1 Δ are the same as in wild-type although the activity is only about 70% of that of wild-type cells. In Figure 5 again the GR level in Y606E hch1 Δ cells seems to be comparable to the wild-type although the activity is reduced to about 20%

3. The authors show that the defects of the W585T mutant can be rescued by additionally mutating A107 to N. They claim that Hsp90 W585T is present in a more open conformation as shown by IP experiments using Sba1 binding as a read-out for the conformational status of Hsp90.

To strengthen the data and to provide evidence that the additional mutation leads to a more closed state of Hsp90, the sensor experiment should be performed with A107N and A107N W585T.

4. The defects of Y606E should be restored in the A107N Y606E variant.

5. Furthermore, it would strengthen the author's model to test for other mutants that show a more closed Hsp90 conformations, e.g. T22I, whether they are able to rescue the W585 and Y606E mutations.

Reviewer #3 (Remarks to the Author):

The studies detailed in the manuscript entitled, "An Hsp90 cochaperone protein in yeast is functionally replaced by site-specific posttranslational modification in humans" takes full advantage of the yeast model system to make a number of important and novel discoveries. The data strongly support a role for yeast Hch1 in regulating yeast Hsp90 conformational dynamics. These findings are of high interest to scientists within the field. The findings that are of more broad interest to science in general and, perhaps, more significant are those that strongly suggest the evolutionary, functional replacement of the Hch1 protein with an Hsp90 PTM in higher eukaryotes. I am not sure if this is the first example of the functional replacement of a protein with a PTM but, if not, it is likely to be one example of only a few. This has far reaching implications regarding how we think about the evolution of protein interaction networks and signaling pathways. The manuscript is well-written and is easy to read. The reader can get a little bit lost in trying to keep track of all of the different mutations and their implications between experiments. I am not sure if there is, but is it possible to provide some type of table or figure the reader could use as somewhat of a quick reference guide to refer to while reading the paper? Overall, the experiments performed and data provided are very thorough and support the conclusions. The authors have also been very careful with their wording in that they have made suggestions in some cases where a firm conclusion cannot be made.

We thank you and the reviewers for taking the time and effort to thoroughly review our manuscript and for the positive comments and suggestions. These have been very helpful in allowing us to strengthen our study and to provide additional mechanistic support for our hypothesis. We have carefully considered each of the suggestions of the reviewers and we detail our responses below (in blue).

Response to reviewers' comments

Reviewer #1 (Remarks to the Author):

- There are two main isoforms of Hsp90 in human, alpha and beta. The authors only examine Hsp90 alpha, which they label Hsp90 in this study. The manuscript should be adjusted to clarify this.

In the modified version of the manuscript we have identified the Hsp90 isoform (Hsp90 α in human cells or Hsp82 in yeast) used in our experiments. We should note that our yeast studies focus on Hsp82 and not Hsc82. Hsp82 is the stress-inducible Hsp90 isoform, as is Hsp90 α in mammals. Both Hsc82 and Hsp90 β are constitutively expressed isoforms in yeast and mammals respectively, and these were not examined in this study (except in response to the reviewer comment below: see our response for additional details).

- Given studies showing differential binding of clients for Hsp90 isoforms, it would be interesting to see the effect of mutating the equivalent residue in Hsp90 beta (Y619). It would be simple to repeat the experiment in Figure 3 using tagged Hsp90 beta.

Following the reviewer's reasonable suggestion, we expressed Hsp90 β -Y619F and E mutants in HEK293 cells and found that, in contrast to the equivalent Hsp90 α phosphomimetic mutant (Y627E), Hsp90 β -Y619E mutation did not impact kinase client (c-Raf & Cdk4 were examined) interaction compared to either wild type Hsp90 β or Hsp90 β -Y619F. These data are intriguing to us and raise some interesting questions that we are currently pursuing (although these experiments are beyond the scope of the current study and will form the basis of a follow-up report). Importantly, a recent study demonstrated that Aha1, whose middle domain binding site on Hsp82 is also used by Hch1, interacts preferentially with Hsp90 α in HEK293 cells (Synoradzki and Bieganowski, *Biochim. Biophys. Acta.* 2015; 1853: 445-452). The authors trace this discrepancy between Hsp90 α and β to the middle domain binding region. Further, these authors pointed out that the C-domain dimerization potential of Hsp90 α and β (Hsp90 α homodimers are significantly more stable compared to homodimers of Hsp90 β) is strongly impacted by a single amino acid difference (in an otherwise highly homologous region of the protein) that is two residues removed from Y627 (Hsp90 α numbering; Y619 in Hsp90 β) and is in the same loop. In Hsp90 α , amino acid 629 is alanine, while amino acid 621 in Hsp90 β is methionine. Replacing alanine with methionine at position 629 in Hsp90 α weakens homodimerization, while replacing the methionine at position 621 in Hsp90 β with alanine strengthens Hsp90 β dimerization. Similar results were also reported by Kobayakawa et al. (*Cell Stress Chaperones.* 2008; 13:97-104). Further, we have preliminary data obtained using molecular homology modeling that this loop region is less mobile, less responsive to N-domain

conformation and more compact in Hsp90 β than it is in Hsp90 α , further highlighting a discrepancy between the two Hsp90 isoforms in this region. We hope to clarify the structural and mechanistic bases of these differences in future experiments and we again would like to thank the reviewer, whose suggestion to examine Y619 mutation in Hsp90 β was the catalyst for initiating this new experimental direction.

- The impact of Hch1 on the W585T has already been examined in Armstrong et al., 2012. The data in figure 1 do not show anything new about this regulation and should be at the very least placed into supplemental data, if not entirely removed.

Although the Armstrong paper did focus on the impact of *HCH1* expression on an Hsp82 mutation, they did not assess its effect on W585T, but rather on A587T. The A587T mutation is located in a similar region within Hsp82 as W585T and this initiated our interest in looking into the impact of *HCH1* expression with *hsp82-W585T*.

- The altered interactions of Cdk4 and Raf-1 have already been published by Xu et al., 2013 and add nothing new to the mechanism of action.

Although Xu et al. (Mol Cell. 2012;47:434-43) did examine the impact of Hsp90 α -Y627E in regard to Cdk4 interaction, the impact of this mutation on Raf-1, GR and v-Src interaction was not examined. The lack of impact seen in this study on GR interaction is novel and highlights the increased complexity of client interactions with mammalian Hsp90 compared to the equivalent Hsp90 isoform in yeast. Additionally, our experiments are focused on the importance of Hsp90 α -Y627 posttranslational modification as a possible functional replacement of a non-conserved yeast co-chaperone; the findings reported in the current manuscript provide the first evidence of such a possibility.

- There is a discrepancy in the blots shown in Figure 3a. In the control lane in 3a, there is a GR band missing in the lysate blots. In 3b, showing a similar experiment, there is a GR band in the control lysate lane. How do the authors explain this?

We thank the reviewer for pointing this out; this was due to a cropping and labeling error, which has been corrected.

- The authors demonstrate altered binding of Cdk4/Raf-1, but not GR. With only a few examples, it is hard to assess the impact of Y627 phosphorylation on client binding. The authors need to show present more examples using IPs of Hsp90 against defined Hsp90 interactors such as Ulk1, ER and c-Src.

In this study, we demonstrate the impact of Hsp90 α -Y627 posttranslational modification on Hsp90 interaction with four well-known client proteins (v-Src, Cdk4, c-Raf, GR), three of which display altered interaction. We believe these data are sufficient to suggest an impact on client interaction, although, as noted above, the 3 kinases behave differently from the nuclear receptor when comparing the yeast data with the data from human cells. Furthermore, c-Src interaction with Hsp90 (in distinct contrast to v-Src) is very weak and difficult to detect by co-immunoprecipitation. Lastly, since it is another nuclear receptor, ER binds to Hsp90 in a similar

manner as GR; therefore, its interaction in HEK293 cells is unlikely to differ between Hsp90 α -WT and Hsp90 α -Y627E.

- Global proteomics studies of Hch1 interactors (similar to studies as Echtenkamp et al., 2011 and Sun et al., 2015) would provide additional insight into the role of Hch1 in the cell and substrate specificity.

We also believe that a global proteomic study of Hch1 interactors would be interesting to obtain. However, we do not believe that such an analysis fits the scope and aim of the current study, which focuses on the impact of Hch1 and Hsp82 mutations on Hsp82 conformation and chaperone activity, rather than on Hch1 interactors (other than Hsp82).

- Given the conclusion of the paper that the need for Y627 phosphorylation in yeast bypassed via Hch1, it would be interesting to examine the effects of expressing yeast Hch1 in mammalian cells. Expression of Hch1 in mammalian cells should phenocopy Y627E (i.e. altered co-chaperone and client binding).

Thank you for this interesting suggestion. We successfully expressed HA-tagged Hch1 in human (HEK293A) cells. Unfortunately, Hch1 failed to interact with endogenous human Hsp90 (see below). This outcome is not particularly surprising, as yeast Hsp90 is not able to compensate for human Hsp90 in human cells. It is also consistent with our hypothesis that the co-temporal loss of Hch1 and the appearance of Hsp90-Y627 phosphorylation is more than just coincidental.

- In 6B, W585T cells demonstrate sensitivity to Radicicol. Does deletion of Hch1 restore growth?

The deletion of *HCH1* does restore growth in the presence of radicicol, although full restoration does not occur in the presence of higher concentrations of drug. These data are in the original manuscript and can be found in Figure S4.

- The authors use genetic analysis to suggest Y627 phosphorylation alters Hsp90 dimerization. Hsp90 dimerization should be examined through more direct biochemical methods including co-IP of differentially tagged monomers as in Mollapour et al. 2015.

Thank you for this suggestion. Although there are different methods to examine Hsp90 N-domain dimerization, such as FRET and differentially tagged monomers, data that are now well-

accepted by the field and published by a number of laboratories over recent years have demonstrated that Sba1 (p23) interaction with Hsp90 only occurs stably when the N-terminal domains of Hsp90 are dimerized, generating a conformation referred to as 'closed state 2'. In this conformation, which is stabilized by Sba1, Hsp90 ATPase activity is inhibited but the chaperone is poised for ATP hydrolysis. These publications include Li et al. *Biochem Biophys Acta*. 2012;1823:624-635, Li et al. *Nat Struct Mol Biol*. 2013; 20:326-331, Ratzke et al. *Nat Commun*. 2014;107:16101-16106, and Zierer et al. *Nat Struct Mol Biol*. 2016;23:1020-1028. Based on the cumulative findings in these and other papers, Sba1 binding to Hsp82 in yeast is now considered to be an excellent interrogator of closed state 2 occupancy in a cellular environment. Thus, we believe that assessment of nucleotide-dependent Sba1 interaction with Hsp82 in yeast (or co-precipitation of p23 with Hsp90 in human cells) provides strong biochemical evidence demonstrating that stable N-domain dimerization (e.g., closed state 2) has been achieved. That is why we included the data in original Figure 6D (now Figure 7B) showing that Sba1 co-precipitated with wild-type Hsp82 and Hsp82-Y606F, but not with Hsp82-Y606E (Y606 in yeast Hsp82 is equivalent to Y627 in human Hsp90 α). Likewise, in Figure S4, we show that endogenous human p23 co-immunoprecipitates with wild-type Hsp90 α and with Hsp90 α -Y627F, but markedly less so with Hsp90 α -Y627E in HEK293 cells. Conversely, in Figure 6C we showed that increased expression of Hch1 dramatically reduced Sba1 association with wild-type Hsp82. In the revised manuscript, we have extended this experimental approach (see Figure 7C) to show that inclusion of A107N mutation (which itself favors N-domain dimerization by promoting ATP lid closure) rescues Sba1 binding to Hsp82-Y606E, much as it rescues Sba1 binding to Hsp82-W585T (see Figure 6D).

Reviewer #2 (Remarks to the Author):

1. Hch1 was previously described as a weak stimulator of Hsp90's ATPase activity – this seems to be in conflict with the authors' model, that Hch1 keeps Hsp90 in an open conformation. Direct evidence of the open state in the presence of Hch1 would be interesting.

Hch1 may weakly stimulate the ATPase activity of Hsp90 by impacting its conformation. Our data in this manuscript show that overexpression of Hch1 reduces the level of Sba1 interaction with Hsp90 (see Figure 6C in revised manuscript), thus relieving the ATPase inhibitory activity of Sba1 bound to Hsp82. Dissociation of Sba1 from Hsp82 would be expected to cause a modest apparent stimulation of ATPase activity, as it allows the stalled ATPase-competent Hsp82 protein to hydrolyze the bound ATP. Indeed, we believe that Hch1 normally comes into play at the end of the Hsp90 ATPase cycle to promote client release and to allow Hsp90 to return to an open conformation and begin a new chaperone cycle.

2. The authors mention in the text that the GR and Ste11 Δ N expression level corresponded to the level of activity. In Figure 2 this is only true for Ste11 Δ N since GR levels in the W585T hch1 Δ are the same as in wild-type although the activity is only about 70% of that of wild-type cells. In Figure 5 again the GR level in Y606E hch1 Δ cells seems to be comparable to the wild-type although the activity is reduced to about 20%.

Thank you for pointing this out; we have modified our discussion of this point in the revised text

to state that there is a variable correlation of client activity with its protein level.

3. The authors show that the defects of the W585T mutant can be rescued by additionally mutating A107 to N. They claim that Hsp90 W585T is present in a more open conformation as shown by IP experiments using Sba1 binding as a read-out for the conformational status of Hsp90. To strengthen the data and to provide evidence that the additional mutation leads to a more closed state of Hsp90, the sensor experiment should be performed with A107N and A107N W585T.

Thank you for this suggestion. We believe the data from this experiment added valuable information to the manuscript and provided additional mechanistic support for our hypothesis. These data (see Figure 6D of revised manuscript) demonstrate that the A107N mutation alone dramatically increases Sba1 interaction with Hsp82 in the presence of nucleotide; this agrees with previous data that demonstrated the A107N mutation favors the closed conformation (see above). Furthermore, addition of this mutation to Hsp82-W585T restored its nucleotide-dependent interaction with Sba1, confirming that the conformational defect caused by W585T is rescued by this secondary mutation favoring stabilization of the closed conformation.

4. The defects of Y606E should be restored in the A107N Y606E variant.

To determine whether addition of A107N to Hsp82-Y606E would similarly restore its chaperoning defect, we monitored GR activity in the presence of *HSP82* WT, A107N, Y606E and A107N-Y606E. Our data show that the functional defect caused by Y606E is partially relieved by inclusion of the A107N mutation. Partial rescue is consistent with the observation that Y606E mutation has a more severe impact on client activity and cell growth than does *HCH1* overexpression. Further, as discussed in reply to reviewer #1, inclusion of A107N mutation in Hsp82-Y606E markedly restored Sba1 interaction (by more than 6-fold compared to Hsp82-Y606E itself). Taken together, these findings are internally consistent and support our original hypothesis.

5. Furthermore, it would strengthen the author's model to test for other mutants that show a more closed Hsp90 conformations, e.g. T22I, whether they are able to rescue the W585 and Y606E mutations.

We assessed the ability of W585T and Y606E mutations to rescue the T22I and T22E phenotypes; however, neither mutation was able to rescue their defects. We believe these findings can be explained by the fact that T22I mutation enhances Hsp82 ATPase activity but in a manner distinct from A107N mutation. Unlike A107N, T22I does not promote stable occupancy of closed state 2 (at which point ATPase activity is inhibited due to increased Sba1 interaction). In support of this possibility, Mollapour et al. (Mol Cell. 2011;41:672-681) reported that T22E mutation did not alter Hsp82 interaction with Sba1 (compared to WT Hsp82), further suggesting that T22I or E mutation does not affect Hsp82 occupancy of closed state 2, unlike A107N mutation.

Reviewer #3 (Remarks to the Author):

The studies detailed in the manuscript entitled, "An Hsp90 cochaperone protein in yeast is functionally replaced by site-specific posttranslational modification in humans" takes full advantage of the yeast model system to make a number of important and novel discoveries. The data strongly support a role for yeast Hch1 in regulating yeast Hsp90 conformational dynamics. These findings are of high interest to scientists within the field. The findings that are of more broad interest to science in general and, perhaps, more significant are those that strongly suggest the evolutionary, functional replacement of the Hch1 protein with an Hsp90 PTM in higher eukaryotes. I am not sure if this is the first example of the functional replacement of a protein with a PTM but, if not, it is likely to be one example of only a few. This has far reaching implications regarding how we think about the evolution of protein interaction networks and signaling pathways. The manuscript is well-written and is easy to read. The reader can get a little bit lost in trying to keep track of all of the different mutations and their implications between experiments. I am not sure if there is, but is it possible to provide some type of table or figure the reader could use as somewhat of a quick reference guide to refer to while reading the paper? Overall, the experiments performed and data provided are very thorough and support the conclusions. The authors have also been very careful with their wording in that they have made suggestions in some cases where a firm conclusion cannot be made.

We appreciate the reviewer's positive comments and the suggestion to provide a table clarifying the purpose/functional impact of each mutation used in this study. We have now included a supplemental table (S3) in the revised Supplemental Information that lists, describes and references the impact of each mutation on Hsp90 conformation and function.

Once again, we thank the reviewers for their thoughtful comments and suggestions and we appreciate the opportunity to submit a revised version of our manuscript incorporating many of their helpful suggestions.

REVIEWERS' COMMENTS:

Reviewer #1 (Remarks to the Author):

The authors have addressed all my previous concerns and in doing so have increased the depth and novelty of an already fascinating story. I enthusiastically support the publication of this manuscript.

Andrew Truman Ph.D.

Reviewer #2 (Remarks to the Author):

The authors have addressed all my queries and resolved open issues.